# A Scoping Review of Epidemiological, Ergonomic, and Longitudinal Cohort Studies Examining the Links between Stair and Bathroom Falls and the Built Environment

**DOI:** 10.3390/ijerph16091598

**Published:** 2019-05-07

**Authors:** Nancy Edwards, Joshun Dulai, Alvi Rahman

**Affiliations:** 1School of Nursing, University of Ottawa, Ottawa, ON K1S 5L5, Canada; jdulai2@uottawa.ca; 2Department of Epidemiology, Biostatistics, and Occupational Health, McGill University, Montreal, QC H3A 0G4, Canada; alvi.rahman@mail.mcgill.ca

**Keywords:** epidemiological studies, ergonomic studies, longitudinal studies on aging, falls, environmental hazards, building codes

## Abstract

Stair and bathroom falls contribute to injuries among older adults. This review examined which features of stairs and bathrooms have been assessed in epidemiological, ergonomic, and national aging studies on falls or their risk factors. Epidemiological and ergonomic studies were eligible if published from 2006–2017, written in English, included older persons, and reported built environment measures. The data extracted included the following: study population and design, outcome measures, and stair and bathroom features. National aging studies were eligible if English questionnaires were available, and if data were collected within the last 10 years. Sample characteristics; data collection methods; and data about falls, the environment, and assistive device use were extracted. There were 114 eligible articles assessed—38 epidemiologic and 76 ergonomic. Among epidemiological studies, 2 assessed stair falls only, 4 assessed bathroom falls only, and 32 assessed falls in both locations. Among ergonomic studies, 67 simulated stairs and 9 simulated bathrooms. Specific environmental features were described in 14 (36.8%) epidemiological studies and 73 (96%) ergonomic studies. Thirteen national aging studies were identified—four had stair data and six had bathroom data. Most epidemiologic and national aging studies did not include specific measures of stairs or bathrooms; the built environment descriptions in ergonomic studies were more detailed. More consistent and detailed environmental measures in epidemiologic and national aging studies would better inform fall prevention approaches targeting the built environment.

## 1. Introduction

Falls are a major threat to the health and independence of community-living older adults. They are the leading cause of injury-related hospitalizations among seniors [1,2,3], and contribute substantially to health care costs [4,5,6], accounting for an estimated $31 billion annually in the United States alone [6]. Indoor and outdoor stairs and bathrooms are locations of particular concern, because they are associated with a substantially higher proportion of injurious falls than those that occur in other locations [7]. 

Given the magnitude of falls and their costs to society, it is evident that population-based approaches are essential in order to make a significant dent in this problem [8,9]. Proposing changes to building codes is one such approach, but robust data, showing causal associations between falls and specific features of the built environment are needed to support and inform such legislative changes [10].

Building codes contain very precise technical specifications. Thus, any proposal for code changes must be accompanied by a justification for and details of the new technical requirements. This scoping review was prompted by the experience of the first author as both a member and observer of technical review committees for the Canadian building codes, and as a proponent of a code-change for universal access to bathtub grab bars. A recurring question from committee members concerned the strength and specificity of the evidence linking features of the built environment (and in particular stairs, guards and bathroom grab bars) to the risk of falls. We could find no systematic reviews on this topic, but examined some of the ergonomic and epidemiological studies that had been presented to Canadian technical committees as research evidence. A preliminary review of these ergonomic studies indicated that they provided strong evidence of how people (and often younger populations) made biomechanical adjustments to navigate simulated environmental changes, such as the steepness of a stair run. However, the evidence from the epidemiological studies was problematic, as it failed to capture the causal links between the specific features of the built environment that must be stipulated in building codes (e.g., diameter of handrails and tread length on stairs) and the incidence and health consequences of falls. Therefore, this review was undertaken to identify and examine the published epidemiological and ergonomic studies pertinent to this topic. We aimed to answer the following questions:What specific environmental features and dimensions of stairs and bathrooms have been assessed in epidemiological studies examining risk factors for falls and/or fall-related morbidity or mortality among seniors in community settings?What specific environmental features and dimensions of stairs and bathrooms have been assessed in ergonomic studies examining the kinematic measures associated with the risk of falling?How does the data collected on the environmental features and dimensions of stairs and bathrooms differ between epidemiological and ergonomic studies?Do national longitudinal studies on aging include questions related to falls, the location of falls, and features of the built environment?

## 2. Materials and Methods

### 2.1. Epidemiological and Ergonomic Studies

We used pertinent Preferred Reporting Items for Systematic Reviews and Meta-Analyses (PRISMA) guidelines [11] for the selection and review of epidemiological and ergonomic studies.

#### 2.1.1. Search Strategy

To identify the published epidemiological and ergonomic studies, we conducted a literature search in the MEDLINE database on the Ovid platform. The search terms were categorized into four concept groups, namely: population, event, outcome, and setting. The specific search terms used and the search strategy are described in Appendix A.

#### 2.1.2. Eligibility Criteria

Eligible studies were published in English between 2006 and 2017, and included some elderly (65 years or older) participants in their sample (however it was not necessary for studies to consist exclusively of elderly participants). The articles had to report primary studies, but could include secondary analyses of the said studies. Studies were ineligible if they were commentaries, review articles, or study protocols with no results. We excluded case studies and case series, as well as studies that used an exclusively qualitative design. All of the other quantitative study designs, such as randomized controlled trials and cohort studies, were included. Studies that reported a sole objective of testing the validity or diagnostic accuracy of a measure were excluded. Studies that only included subjects because they used multifocal glasses, had undergone hip or knee arthroplasty, or were amputees using prosthetics were also ineligible. Studies had to include some information about the built environment for either bathrooms or stairs. We defined stairs as two or more steps. Therefore, studies that only assessed curbs, single steps, or ramps, were excluded. 

There were some differences in the eligibility criteria used for the epidemiological and ergonomic studies, reflecting the different types of questions posed and the related methods used in these two classes of studies. These class-specific criteria are outlined in the next two sections.

#### 2.1.3. Additional Inclusion Criteria for Epidemiological Studies

We included all of the epidemiological studies with 30 or more participants that investigated stair and/or bathroom environmental risk factors for falls, fall-related deaths, or fall-related injuries (including specific sub-types of fall injuries (e.g., cranio-facial injuries)). The use of a fall risk index was not considered a measure of falls. Studies had to include one or more measures of the built environment, defined as a report by the participant of the location of falls (stairs and/or bathroom) and/or specific features of the stairs or bathrooms assessed directly by the interviewer (e.g., bathtub grab bar present or absent, or stair dimensions). Studies that only assessed the participants’ perceptions or knowledge of the environmental risk factors were excluded, as were studies that only assessed participants’ perceptions or self-reports of whether or not they could ascend and/or descend stairs. Similarly, administration of the Stair Climb Test or Power Test [12] alone, did not meet this built environment criterion. Studies examining only risk factors for falls (but with no measure of falls) or only the presence or absence of environmental hazards (but with no measure of falls) were excluded. Studies conducted entirely in inpatient hospital or emergency departments or residential care (long-term care) settings were ineligible as well.

#### 2.1.4. Additional Inclusion Criteria for Ergonomic Studies

Ergonomic studies were included if they were conducted in simulated lab conditions and/or in community settings, where at least ten or more participants had to navigate either a set of stairs or a bathroom configuration (e.g., bathtub and grab bars, or toilet and grab bars). Ergonomic studies had to include one or more kinematic measures directly associated with the likelihood of a fall. Studies that used only virtual reality to simulate environmental conditions were excluded.

#### 2.1.5. Study Selection Process

The literature search results were managed in Zotero [13]. Duplicate articles were removed. The inclusion and exclusion criteria were pilot-tested on a subset of approximately 100 titles and abstracts with independent reviews, by A.R., N.E., and J.D. The results were compared. The criteria were then clarified and finalized. Using the final set of eligibility criteria, two reviewers (J.D. and N.E.) independently conducted title and abstract screenings of 146 retrieved articles; conflicts were resolved by consensus. The remaining 556 articles were screened by J.D. (reviewed 201) and N.E. (reviewed 355). At this level of screening, articles were classified as “yes—eligible”, “no—ineligible”, or “unclear”. Articles that were classified as “yes” were included for data extraction. Articles classified as “unclear” were processed for full-text review. A final screening decision for the “unclear” articles was then made during the full-text screening. When there was uncertainty, two reviewers (J.D. and N.E.) independently reviewed the full text and came to a consensus decision. Finally, the eligibility of all of the articles was reconfirmed during the data extraction process, with some articles deemed ineligible at this stage.

#### 2.1.6. Data Collection

A data extraction form was developed and pilot-tested by all of the authors on a subset of included studies, so as to identify irrelevant and missing fields. The revised data extraction form was then used by one reviewer (JD) for the remaining studies. The following information was collected from the included studies, where relevant:Study information (author(s) and year of publication);Study setting (country for both epidemiological studies and ergonomic studies, and laboratory versus community setting for ergonomic studies);Study characteristics (type of study design, study aims, sample size, and age of study participants);Outcomes for epidemiological studies, namely: data source for falls; measures of falls reported (falls, fall-related injuries, and fall-related deaths); location of falls described (proportion of falls on stairs, in bathrooms or in another location, or location of falls not disaggregated; indoor versus outdoor falls);Outcomes for ergonomic studies, namely: type of kinematic and kinetic variables assessed (e.g., gait, balance, and muscle strength); included question(s) about falls in their study (yes or no);Environmental features for epidemiological studies, namely: data source(s) used for environmental hazards; environmental features and dimensions of stairs and bathrooms recorded (yes or no), and if recorded, the features and dimensions described; individual versus cumulative set of environmental hazards used in analysis of relationship to measure(s) of falls; and,Environmental features for ergonomic studies, namely: environmental features and dimensions of stairs and bathrooms recorded (yes or no), and if recorded, features and dimensions described.

### 2.2. Longitudinal Studies on Aging

For this segment of the review, we first identified national longitudinal studies on aging. We then retrieved and extracted information about the measures used in the questionnaires.

#### 2.2.1. Search Strategy

National longitudinal studies on aging were predominately found by entering the terms “longitudinal study on aging”, “longitudinal studies on aging”, and “national longitudinal studies on aging” into the search engines Google and Google Scholar. The websites of the organizations Gateway to Global Aging Data [14], iLifespan [15], and the Integrative Analysis of Longitudinal Studies of Aging and Dementia (IALSA) [16], located at Maelstrom, were additionally used to locate studies.

#### 2.2.2. Eligibility

Longitudinal studies on aging had to be national in a scope; include persons over the age of 65, although they could also include younger individuals; report the launch of, or ongoing conduct of data collection in the past ten years; include plans for more than one period of data collection; and, have a full English version of the questionnaire retrievable on a website or in a publication.

#### 2.2.3. Data Extraction

The data extracted from the longitudinal study questionnaires and/or code books included the following: study name and time period; sample characteristics; data collection methods (in home or over the phone); and data collected about falls, the built environment, and/or assistive devices used by or accessible to the individual. For the data collected, we determined whether qualitative and/or quantitative data were elicited, what specific questions were asked (including probes), and what response options (if any) were provided.

## 3. Results

### 3.1. Eligible Epidemiological and Ergonomic Studies

The literature search yielded 746 ergonomic and epidemiological articles, with 702 remaining after duplicates were removed (see Figure 1). Following title and abstract screening, 120 articles were included for data extraction, 498 articles were excluded, and 85 articles were classified as “unclear”. After reading full-texts of the 85 “unclear” studies, 31 were included for data extraction and the remaining 54 were excluded.

While conducting the data extraction, we identified an additional 37 studies that did not qualify for inclusion. Therefore, a total of 114 out of 702 studies retrieved in the search were included in the review, and the remaining 587 were excluded. Of these 114 included studies, 38 were epidemiological and 76 were ergonomic.

### 3.2. Characteristics of Epidemiological and Ergonomic Studies

The sample sizes for the ergonomic studies ranged from 10 to 513. Of the 76 ergonomic studies, 67 were performed on simulated stairs and 9 in a simulated bathroom. Only four (5.2%) of the ergonomic studies asked questions about falls.

The majority of epidemiological studies were cross-sectional. Sample sizes for epidemiological studies ranged from 89 to ~3.4 million. Among the 38 epidemiological studies, two reported stair falls only, four reported bathroom falls only, and the remainder (n = 32; 84.2%) examined falls on stairs and in bathrooms, and in some instances in other locations as well. Participant interviews were the most common source of data on falls (n = 20; 52.6%), followed by administrative data, including injury surveillance data (n = 12; 31.6%). Similar data sources were reported for environmental features, namely: participant interviews (n = 17; 44.7%) and administrative data (n = 12; 31.6%). Environmental checklists, self-administered questionnaires, and mortality data were used in a minority of studies (three in each category). Indoor and outdoor locations for falls on stairs were reported separately in only 10 (33.3%) of the studies describing stair falls.

### 3.3. Features of Stairs and Bathrooms Described in Epidemiological and Ergonomic Studies

#### 3.3.1. Epidemiological Studies

Specific environmental features (other than fall location) were examined in 14/38 (36.8%) epidemiological studies (See Table 1). Nine (64.3%) of these 14 studies provided some description of the structural features of the stairs, and 10 (71.4%) provided some description of the environmental features of the bathrooms. Structural descriptions of the stairs included whether or not there was adequate lighting; if a handrail was present; the length, width, and height of the steps or staircase; the number of steps; and whether or not the steps were even. Structural descriptions of the bathrooms included whether or not there was adequate lighting, if there were grab bars in the bathtub or shower, and if there were grab bars near the toilet. Non-structural bathroom features included the presence of non-slip bathmats or wobbly toilet seats.

#### 3.3.2. Association between Falls and Features of Stairs and/or Bathrooms in Epidemiological Studies

Twenty-four studies provided disaggregated falls data on stairs, and 17 studies provided disaggregated falls data in bathrooms. However, one or more features of the stairs that may have contributed to the risk of falling were described in only 8 (33.3%) of the 24 stair falls studies. Similarly, one or more features of the bathroom environments that may have contributed to the risk of falling were described in only 10 (58.8%) of the 17 bathroom falls studies. Furthermore, a number of these studies presented only cumulative environmental risk factor scores (aggregating two or more environmental features), rather than location-specific, fall incidence rates in their analysis. For instance, in a randomized controlled trial, Keall et al. [20] reported a reduction in the overall incidence of falls following home modifications, including modifications to bathrooms and/or stairs. However, their analysis did not examine the association between the specific modifications made and where the falls occurred. Leclerc et al. [26] indicated what proportion of falls occurred in the bathrooms, but their analysis only examined the relationships between the falls and a cumulative environmental risk factor score (combining hazards in bathrooms and other locations in the home). A cross-sectional study by Edwards et al. [30] assessed the presence of bathtub and toilet grab bars in seniors’ apartments, but falls in any location in the past year were used as an outcome variable. Similarly, Lim and Sung [24], and Pereira et al. [18] measured specific bathroom hazards, but used falls in any location as their outcome variable.

There were other ambiguities reflecting the limitations of the data sources. Hensbroek et al. [28] used an injury surveillance database to determine the height of the participants’ stair falls by recording the number of steps they fell from, but the specific structural hazards involved (e.g., top-of-flight fault [31]) were not reported. Lim and Sung [24] identified two significant risk factors associated with falls (presence of a door sill and poor night light), although it was not clear whether these were hazards in the bathroom. Hanba et al. [17] used information from the National Electronic Injury Surveillance System (NEISS) to ascertain locations in bathrooms that may have caused craniofacial traumas in patients, but consistent probes about the absence or presence of grab bars were not used to ascertain this information.

#### 3.3.3. Ergonomic Studies

The majority of stair studies (65/67; 97.0%) described the specific stair dimensions used in their experimental settings, including information such as the number of steps, height of risers, length, and width of treads, height of staircase, and/or degree of incline of staircase (see Table 2).

Four (6%) of these studies also provided the dimensions of the landings for their stairs [45,61,79,94]. Forty-five percent (30/67) described whether or not the stairs had handrails and whether or not participants were instructed or permitted to use the handrails during stair ascent or descent. However, only 13.3% (4/30) of these studies provided detailed information about the handrail dimensions, such as their height and diameter [47,54,61,95]. Foster et al. [48] and Foster et al. [56] examined the use of a tread edge highlighter on a set of stairs, while three other studies simply described their presence on the staircase [79,94,95], but without testing the tread edge marking as an experimental condition. A few ergonomic studies provided other details about the study setting such as the lighting conditions [77,79,95] or the material or pattern of the coverings on the stairs [48,77,95].

With one exception, all of the ergonomic studies of bathrooms provided detailed information about some features of the bathroom setting. King and Novak [33] evaluated the effectiveness of five different bathtub assistive device conditions (vertical grab bar, horizontal grab bar, bath mat, sidewall, or no assistance) on balance during bathing transfers. Guitard et al. [71] provided details of four different grab bar conditions, namely: no grab bars, vertical and horizontal combination, L-shaped bar, and vertical/angled combination. Kennedy et al. [49] evaluated six different grab bar configurations for toilets (commode, two vertical bars, one vertical bar, one horizontal bar, swing-away bars, and one diagonal bar). Another study, focusing on commodes, utilized sensors to examine grab bar usage among participants during stand-to-sit and sit-to-stand transfers [85]. 

Kinematic measures related to the risk of falling that were commonly assessed in ergonomic studies were gait analysis, balance, and muscle strength. Only four studies reported whether falls occurred during the simulation [34,77,84,104]; participants fell in only one of these studies [104]. Fall history was only asked in three studies [64,71,98], with one of these studies following up with participants after the lab simulation to ask about falls [64].

### 3.4. National Longitudinal Studies on Ageing

#### Studies Retrieved and Study Characteristics

The questionnaires for most of the longitudinal studies were found on the studies’ websites and/or from one of the three websites used to locate the national longitudinal studies on aging. In one instance, the questionnaire information was obtained from a study’s journal publications and a coding manual. Although 18 longitudinal studies on aging were identified, English versions of the questionnaires could only be located for 13 of these studies. The characteristics of these 13 studies are listed in Table 3.

All of the questionnaires included items about falls, but with considerable variability in the information collected. Variations included the time period since the fall occurred (six months to two years), and whether or not questions were asked about the location of falls (two studies asked about location) or about the activities being undertaken when the fall occurred (three studies asked about activities). The information obtained about the environmental hazards was sparse and uneven across the studies, reflecting both how the question was asked and what response options were given (if any). Only four studies (Canada, Germany, Sweden, and Ireland) included questions about stairs in their surveys. However, only two of these studies asked participants if any of their falls had occurred on the stairs. In the Canadian Longitudinal Study on Aging, participants were asked how their fall happened, with one of the response options being using stairs and steps. Similarly, the Irish Longitudinal Study of Aging asked participants if their fracture was due to a fall, and if so how did the fall happen (one of the response options was using stairs). However, neither of these studies collected data on the actual features of the stairs implicated in the fall. The German and Swedish studies assessed the accessibility and safety of stairs, but this data could not be linked to participants’ reports of falls. The German Ageing Survey assessed whether or not the participants’ homes could be reached without stairs, with ten or fewer steps, or with more than ten steps. They recorded whether or not handrails were on both sides of all of the stairs in the house, and if every room was accessible without steps. Similarly, the Swedish National Study on Aging and Care recorded if participants’ homes were accessible for people with mobility issues, with response options including “elevator and a few steps” and “one floor staircase”. Neither of these studies, however, were able to link these features of stairs to any falls participants may have had. None of the 13 longitudinal studies obtained actual measurements of the stairs, despite the fact that the majority of studies (9/13, 69.2%) conducted portions of the interviews in the participants’ homes, most often to collect physiologic data and/or biological specimens. 

With regards to bathroom modifications, the studies in Canada and Ireland explicitly asked participants if any of their falls had occurred while getting into or out of the bathtub or shower. In addition, longitudinal studies from these two countries as well as those from Costa Rica, England, the United States, and Europe, asked participants if modifications had been made to their bathrooms, such as the installation of grab bars or hand rails. However, the timing of the grab bar installation, in relation to the timing of falls, was not recorded. In addition, other details such as the number of grab bars; the slope, placement, and the height of the grab bars; and whether or not they were permanently installed, were not recorded in any of the studies.

### 3.5. Comparison of Epidemiological, Ergonomic, and Longitudinal Studies on Aging

Overall, data on the specific features of stairs and bathrooms were sparse for the epidemiological studies and more detailed for the ergonomic studies. While nearly all of the national longitudinal studies on aging collected data on both falls and features of the environment, specific falls could only be linked to these environmental attributes in three (23%) of these studies.

## 4. Discussion

Epidemiological and ergonomic studies should provide an important source of evidence to guide improvements in the built environment that may help prevent falls. This is the first review to compare these two types of data, despite long-standing observations suggesting that environmental hazards are implicated in approximately 30%–47% of all falls [119,120,121,122,123,124,125]. Our comparative results suggest some important data gaps and provide the basis for new directions in the study of fall risks, fall prevention, and the built environment. 

The results highlight substantial differences in the types of data collected in epidemiological and ergonomic studies, and therefore, their joint potential to attribute falls to specific environmental features. In order to attribute falls to an environmental feature, the following are required: the location, circumstances, and mechanism of the fall; as well as specific measures of the environmental entity. However, in the majority of epidemiological studies, some or all of this data was either missing, or there was no possibility of linking the occurrence of a particular fall with a specific environmental characteristic. In studies that included specific measures of environmental features and fall location, the analysis of the relationship between the two often used either cumulative incidence scores (falls in all locations) or cumulative environmental hazard scores, precluding any examination of the direct relationship between a specific hazard and a fall involving that hazard. While ergonomic studies provided considerably more details about the stairs and bathrooms, only a minority of these studies included data on the participants’ histories of falls in either of these locations.

It is understandable that earlier epidemiological studies emphasized the location of the falls (indoor versus outdoor or room location in a home), and tried to disentangle clinical, behavioral, and environmental risk factors [126,127,128]. However, it is apparent that the literature on fall risk factors and interventions studies has progressed more along clinical and behavioral avenues of inquiry, and less around the built environment. For example, clinical measures of risk factors for falls have advanced substantially and there is solid consensus on the measures that should be routinely used in clinical assessments. In comparison, our review of epidemiological studies indicates that measures of the built environment are inconsistent across studies.

Only ten epidemiological studies differentiated between falls on indoor versus outdoor stairs, and none asked whether falls occurred on the main stairs or main level of a home or on another staircase (e.g., to a basement), which may involve less stringent building codes. There was also no differentiation between staircases and steps. This is also problematic, because code requirements for staircases (often defined as three or more steps) are often more specific and restrictive than the requirements for one or two steps.

Furthermore, although some interventions to reduce environmental hazards [20,129] have demonstrated effectiveness, the majority have focused on changes that can more readily be made by residents, such as installing handrails on stairs or grab bars in bathrooms, rather than more substantial structural changes that require legislated changes, such as building code improvements to decrease the height of risers and increase the tread length of stairs. Measurement specificity is of particular importance in order to detail the technical requirements of codes. 

Building codes, by design, are universal in their application. Therefore, it was also notable that, with the exception of a study by Edwards et al., [30] none of the epidemiological studies reported whether or not environmental features such as grab bars in bathrooms or handrails on stairs were universally available. This kind of information is needed in order to make the connection between built environment features and related policies. This would allow for a comparison of fall rates in those settings, with and without universal access requirements. This could also help inform building code policies.

Differences in outcomes and a myriad of environmental measures make it nearly impossible to find a way to assemble the literature in this area. This is an important problem, because among those studies measuring falls in any location, only a subset occur on stairs or in bathrooms. While these constitute a considerable proportion of injurious falls, even with a sizable sample of participants, the percentage of stair and bathroom falls is often too small for a sub-group analysis. While standard definitions of falls are in common use [130], standard definitions for environmental hazards are lacking, as we have reported elsewhere [131]. Ideally, with a more standardized approach to environmental measurement, data on stair and bathroom falls could be assembled across studies so as to examine the specific patterns of environmental features, and their relationship to falls and fall-related injuries.

We were surprised to find the gaps, inconsistencies, and disconnects in measuring the falls and environmental features in the longitudinal studies on aging. The lack of environmental features was in sharp contrast to the inclusion of measures on assistive devices, such as canes and walkers, reported in 76.9% of the 13 cohort studies. These large-scale cohorts yield substantial numbers of falls, and these would be adequate either within or between studies for sub-group analyses of stair and bathroom falls. Nevertheless, a lack of data on rather basic information, such as where the fall occurred (absent in 84.6% of the longitudinal studies on aging) precludes such analysis. Given the variable environmental conditions across countries, these national longitudinal studies on aging could provide an important basis to compare different regulated and unregulated environmental features. In our estimation, this could be enormously helpful for building code related decisions. Perhaps it is because building code regulations exist in a non-health sector that limited attention has been paid to them. It is time to bridge these intersectoral disconnects and strengthen cohort studies, so they yield policy relevant data for both health and other sectors.

Those working in the field of ergonomics may argue that epidemiological studies are not needed, as more refined modifications to stairs or bathrooms can be better tested in simulated settings. Nevertheless, there are problems with this exclusive approach. First, ergonomic studies do not enroll representative populations, in fact those with a high likelihood of falling may be excluded from such studies because of ethical concerns (as they may be at risk for an injury even when they are put in a safety harness). Second, while some specific built environment variations are easy to simulate, others are not. The absence of lighting is an example, as it is difficult to simulate the uneven combination of natural and artificial lighting that may be found on a home stairway. Lab stairways often consist of only a few steps; but individuals may need to navigate much longer staircases and/or problematic landings in the real world, where there are additional distractions to overcome. 

As we expected, ergonomic studies that simulated stairs included substantially more detailed information regarding stair characteristics and features, as the simulated laboratory or community set-up was part of the methods’ description, and precise, standardized parameters were provided. However, very few of these studies measured falls and fall outcomes, which are not interpretable from ergonomic studies, given their small sample sizes and only a few tests of stair ascent and descent for each participant. Thus, ergonomic studies do not investigate the association between the specific features of stairs and the risk of falls, making it difficult to ascertain which features should be modified in order to improve the safety of stairs. Nevertheless, it may be beneficial for epidemiologists to use ergonomic studies as a guide to develop a core set of environmental features that should be measured. Our review suggests that ergonomists and epidemiologist have not as yet brought their joint expertise together to address the issue of the built environment and falls. 

It is time to develop a more substantial evidence-base that can inform improvements to building codes and help shape the construction of our homes. Changes to the built environment can help to reduce the risk of falls among seniors. A population health approach to fall prevention involves making changes to building code requirements for the construction of public and private stairs, as well as bathrooms. Research evidence used to inform building code revisions has primarily been in the form of ergonomic or laboratory studies, as they usually contain specific information about stair dimensions or bathroom features (e.g., grab bars). Epidemiological studies, with robust and specific measures of the built environment, are needed in order to strengthen the evidence base for modifications to the built environment.

### 4.1. Limitations

Data extracted from published studies were limited by what variables the authors presented in their findings, but there were no indications that details about environmental variables were missing from the methods’ descriptions. Our 12-year period for eligibility may have excluded earlier studies with more precise measures of the environment. However, this seems unlikely, based on a review of some earlier studies examining risk factors (including environmental hazards) for falls [119,120,121,122,123,124,125,132]. Limiting our search to the MEDLINE database may have excluded some eligible studies. However, the initial yield of our MEDLINE search (702 articles) was substantial and included studies that came from a wide range of countries (n = 24) and journals. An additional confirmatory MEDLINE search, which included non-English articles, yielded no further eligible non-English studies.

Our focus for data extraction was bathrooms and stairs, as these are critical locations for falls, and in particular, injurious falls [9,133]. There are however, other environmental features and related policy implications that should also be taken into account in future research, as communities move towards the development of an age-friendly infrastructure.

### 4.2. Recommendations

Four recommendations emerge from these findings. First, epidemiologists should invite ergonomists to help identify precise measures of the environment that could be included in epidemiological studies. They should also confer with members of technical committees who are tasked with reviewing and proposing improvements to building codes, so as to identify what specific data gaps need to be addressed. 

Second, consensus definitions of falls were developed in the 1980s [134]; these have been instrumental in advancing research in this field, as they provide comparable outcomes measures. Consensus definitions of environmental features are essential in order to advance our understanding of the causal relationships between the built environment and falls. Professionals with ergonomic training have an important role to play in helping to develop such consensus measures. 

Third, longitudinal studies of aging involve major financial investments by governments. In order for these studies to yield data that can better guide improvements in the built environment, more precise environmental measures are needed, and the location of falls must be reported so that data on falls and environmental features can be linked and attribution ascertained. Given the in-home collection of other physiological data for the majority of these longitudinal studies, specific features of environmental hazards could be readily assessed by trained assistants, augmenting seniors’ accounts of what happened during a fall. Longitudinal studies provide an opportunity for the efficient collection of such data, and provide a stronger evidence base than cross-sectional studies to attribute falls to particular features of the built environment.

Finally, given the long-standing differences in the built environment and in building codes in various countries, inter-country comparisons of specific environmental features and their relationship to falls using ecological designs affords an important means to advance this work. Longitudinal studies across settings, purposefully selected on the basis of variable features of the built environment, offer a robust data base for such comparisons.

## 5. Conclusions

This review indicates that epidemiological studies, including longitudinal studies of aging, lack the data required to examine the relationships between specific features of the built environment and falls. Ergonomic studies provide a useful basis to inform more precise measures of the built environment. However, until better measures of the built environment are included in studies of falls by epidemiologists and those conducting longitudinal studies of aging, population health approaches, and, in particular, evidence-informed changes to building code policies will remain in abeyance.

## Figures and Tables

**Figure 1 ijerph-16-01598-f001:**
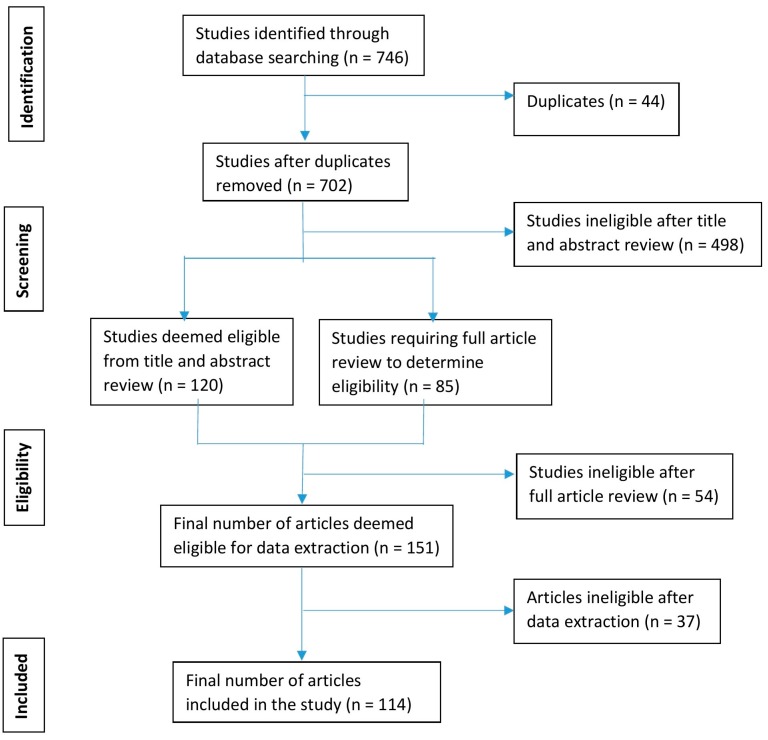
Preferred Reporting Items for Systematic Reviews and Meta-Analyses (PRISMA) flow diagram.

**Table 1 ijerph-16-01598-t001:** Features of stairs and bathrooms described in the epidemiologic studies that assessed falls, fall-related morbidity, and/or fall-related mortality as outcomes.

Articles by Author and Year	Quantitative Measure of Environmental Features of Stairs (n = 9 Stair Studies)							Quantitative Measure of Environmental Features of Bathrooms (n = 10 Bathroom Studies)			
	Lighting	Handrail	Stair Rise or Run Dimensions	Uneven Stairs	Tread Edge Contrast	Landing Size	Other ^1^	Lighting	Shower or Bathtub Grab Bar	Toilet Grab Bar	Other ^2^
Hanba et al. (2017) [17]											✓
Pereira et al. (2017) [18]								✓	✓	✓	✓
Stessman (2017) [19]		✓									
Keall et al. (2015) [20]	✓	✓					✓		✓	✓	✓
Shi et al. (2014) [21]	✓		✓								
Kamel et al. (2013) [22]	✓	✓		✓				✓	✓	✓	✓
Kuhirunyaratn et al. (2013) [23]										✓	
Lim and Sung (2012) [24]											✓
Sophonratanapokin et al. (2012) [25]		✓							✓	✓	✓
Leclerc et al. (2010) [26]	✓	✓	✓	✓			✓	✓	✓		✓
Chan et al. (2009) [27]							✓				✓
Hensbroek et al. (2009) [28]			✓								
Yu et al. (2009) [29]	✓										
Edwards et al. (2006) [30]									✓	✓	✓
**Total N (% of all epidemiological studies)**	**5/38 (13.2%)**	**5/38 (13.2%)**	**3/38 (7.9%)**	**2/38 (5.3%)**	**0/38 ^3^ (0.0%)**	**0/38 ^3^ (0.0%)**	**3/38 (8.8%)**	**3/38 (7.9%)**	**6/38 (15.8%)**	**6/38 (15.8%)**	**9/38 (23.6%)**

^1^ Table only includes 14 studies that assessed one or more specific features of stairs and/or bathrooms. Twenty-four included studies that only identified the location of the falls are not listed here. ^2^ Slip-resistant edging, slip-resistant surface, door sill, steps in need of repair, loose treads or carpeting, or type of floor materials. ^3^ Mats and rugs; unspecified hazards; slippery surfaces; toilet seat (wobbly or too low); bath/shower seat, chair, or bench; or hydraulic lifts.

**Table 2 ijerph-16-01598-t002:** Features of stairs and bathrooms described in ergonomic studies.

Articles by Author and Year	Quantitative Measure of Environmental Features of Stairs							Quantitative Measure of Environmental Features of Bathrooms			
	Lighting	Handrail	Stair Rise or Run Dimensions	Uneven Stairs	Tread Edge Contrast	Landing Size	Other ^1^	Lighting	Shower or Bathtub Grab Bar	Toilet Grab Bar	Other ^2^
Johannsson et al. (2017) [32]			✓								
King and Novak (2017) [33]									✓		
Kunzler et al. (2017) [34]			✓								
Song et al. (2017a) [35]			✓								
Song et al. (2017b) [36]			✓								
Swanenburg et al. (2017) [37]		✓	✓								
Wang et al. (2017) [38]		✓	✓								
Handsaker et al. (2016) [39]			✓								
Lyytinen et al. (2016) [40]			✓								
Novak et al. (2016) [41]		✓	✓								
Shin and Yoo (2016) [42]			✓								
Spolaor et al. (2016) [43]			✓								
Weiss et al. (2016) [44]			✓								
Alcock et al. (2015) [45]			✓			✓					
Brodie et al. (2015) [46]			✓								
Chiu et al. (2015) [47]		✓	✓								
Foster et al. (2015) [48]		✓			✓		✓				
Kennedy et al. (2015) [49]										✓	
Laudanski et al. (2015) [50]			✓								
Paquette et al. (2015) [51]			✓								
Singhal et al. (2015) [52]		✓	✓								
Alcock et al. (2014) [53]		✓	✓								
Antonio and Perry (2014) [54]		✓	✓								
Foster et al. (2014a) [55]			✓								
Foster et al (2014b) [56]			✓		✓						
Handsaker et al. (2014) [57]			✓								
Hinman et al. (2014) [58]		✓	✓								
Hsue and Su (2014) [59]			✓								
Singhal et al. (2014) [60]		✓	✓								
Telonio et al. (2014) [61]		✓	✓			✓					
Wang et al. (2014) [62]			✓								
Buckley et al. (2013) [63]			✓								
Guitard et al. (2013) [64]									✓		
Novak and Brouwer (2013) [65]			✓								
Samuel et al. (2013) [66]		✓	✓								
Bosse et al. (2012) [67]			✓								
Hicks-Little et al. (2012) [68]			✓								
Novak and Brouwer (2012) [69]		✓	✓								
Samuel et al. (2012) [70]		✓	✓								
Guitard et al. (2011) [71]									✓		
Hicks-Little et al. (2011) [72]		✓	✓								
Karamanidis and Arampatzis (2011) [73]		✓	✓								
Leitner et al. (2011) [74]		✓	✓								
Lin et al. (2011) [75]											✓
Novak and Brouwer (2011) [76]			✓								
Oh-Park et al. (2011) [77]	✓	✓	✓				✓				
Reid et al. (2011) [78]		✓	✓								
Zietz et al. (2011) [79]	✓	✓	✓		✓	✓					
Fujita et al. (2010) [80]			✓								
Liikavainio et al. (2010) [81]		✓	✓								
Pua et al. (2010) [82]		✓	✓								
Reid et al. (2010) [83]			✓								
Siegmund et al. (2010) [84]									✓		✓
Arcelus et al. (2009) [85]										✓	
Asay et al. (2009) [86]		✓	✓								
Bertucco and Cesari (2009) [87]			✓								
Hsue and Su (2009) [88]			✓								
Karamanidis and Arampatzis (2009) [89]			✓								
Kim (2009) [90]		✓	✓								
Misic et al. (2009) [91]		✓	✓								
Ojha et al. (2009) [92]			✓								
Reeves et al. (2009) [93]			✓								
Schwartz et al. (2009) [94]		✓	✓		✓						
Zietz and Hollands (2009) [95]	✓	✓	✓		✓		✓				
Di Fabio et al. (2008) [96]		✓	✓								
Larsen et al. (2008) [97]			✓								
Reeves et al. (2008) [98]		✓	✓								
Guo et al. (2007) [99]			✓			✓					
Lee and Chou (2007) [100]			✓								
Liikavainio et al. (2007) [101]			✓								
Mian et al. (2007) [102]			✓								
Misic et al. (2007) [103]		✓	✓								
Murphy et al. (2006) [104]	✓									✓	
**Total N (%)**	**3/67 (4.5%)**	**30/67 (44.8%)**	**64/67 (95.5%)**	**0/67 ^3^ (0.0%)**	**5/67 (7.5%)**	**4/67 (6.0%)**	**3/67 (4.5%)**	**0/9 ^3^ (0.0%)**	**5/9 (55.6%)**	**2/9 (22.2%)**	**2/9 (22.2%)**

^1^ Horizontal vertical illusion on the stairs, non-carpeted stairs, or stair surface material. ^2^ Tiles, anti-slipping mat, or slip resistant bath tub. ^3^ No studies reported on uneven stairs or lighting conditions of bathroom simulations.

**Table 3 ijerph-16-01598-t003:** Features of stairs and bathrooms in national longitudinal aging studies.

Name of Study	Falls in all Locations Assessed	Assistive Device Use Recorded	Quantitative Measure of Environmental Features of Stairs							Quantitative Measure of Environmental Features of Bathrooms			
			Lighting	Handrail	Stair Rise or Run Dimensions	Uneven Stairs	Tread Edge Contrast	Landing Size	Other ^1^	Lighting	Shower or Bathtub Grab Bar	Toilet Grab Bar	Other ^2^
Australian Longitudinal Study of Ageing (1992–2014) [105]	✓	✓											
Canadian Longitudinal Study on Aging (2013–present) [106]	✓	✓											
Costa Rican Longevity and Health Aging Study (2005–present) [107]	✓	✓									✓		✓
English Longitudinal Study of Aging (2002–present) [108]	✓	✓		✓					✓				✓
German Ageing Survey (199–present) [109]	✓	✓		✓					✓				
Health & Retirement Study (USA; 1992–present) [110]	✓	✓		✓					✓		✓		✓
Japanese Study on Aging and Retirement (2007–present) [111]	✓												
Korean Longitudinal Study of Aging (2006–present) [112]	✓												
Mexican Health and Aging Study (2001–present) [113]	✓	✓											
New Zealand Longitudinal Study of Ageing (2006–present) [114]	✓												
Survey of Health, Ageing, and Retirement in Europe (2004–present) [115]	✓	✓		✓					✓				✓
Swedish National Study on Aging and Care in Kungsholmen (2001–present) [116,117]	✓^3^	✓											
The Irish Longitudinal Study on Ageing (2009–present) [118]	✓	✓											✓
**Total N (%)**	**13/13 (100.0%)**	**10/13 (76.9%)**	**0/13 (0.0%)**	**4/13 (30.8%)**	**0/13 (0.0%)**	**0/13 (0.0%)**	**0/13 (0.0%)**	**0/13 (0.0%)**	**4/13 (30.8%)**	**0/13 (0.0%)**	**2/13 (15.4%)**	**0/13 (0.0%)**	**5/13 (38.5%)**

^1^ Door sills or ramps. ^2^ Walk in shower—level access/standard shower tray, bath/shower seat, toilet equipment/commode, stool, apparatus/instrument to use toilet, raised toilet seat, or bathroom aids or grab bars not specified. ^3^ Fall location mentioned in publication [117], but item not listed in questionnaires.

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
