# Peer review of "A Scoping Review of Epidemiological, Ergonomic, and Longitudinal Cohort Studies Examining the Links between Stair and Bathroom Falls and the Built Environment"

_ijerph, 2019, doi:10.3390/ijerph16091598_

Round 1

Reviewer 1 Report

Thanks for the revisions by the authors. I have no more comments on this paper.

Author Response

Please see document attached for response to Reviewer 1.

Reviewer 2 Report

This is an excellent paper and was a pleasure to read.

I only have 1 very minor grammatical comment:

P19 L400-2 "However, very few of these studies measured falls and fall outcomes ??WHICH are not interpretable from ergonomic studies given their small sample sizes and only a few tests of stair ascent and descent for each participant."

Author Response

Please see document attached for response to Reviewer 2.

This manuscript is a resubmission of an earlier submission. The following is a list of the peer review reports and author responses from that submission.

Round 1

Reviewer 1 Report

This paper provides interesting results and is valuable for elderly fall prevention. However, a few issues need to be clarified or improved before it is accepted for publication.

1. Why does this review exclude trial on this topic? In general, randomized, controlled trails are stronger than cohort studies. If possible, I suggest inclued RCTs in the searching scope.  

2.  It is unclear why the authors focus studies at the national level. In fact, researches at local level are also valuable for improving elderly fall prevention.

3. The omission of non-English languages may miss some important studies. This should be mentioned as a limitation.

4. It is unclear why limited the publication time period to the last decade. In most cases, studies that were published ten years ago are also important.

5. It is unclear why the authors did not assess the study quality. I think this is particularly important for evidence evaluation. I recommend to add the information of study quality of the included studies.

6. It is unclear why the authors limited epidemiological studies with 30 participants and more and ergonomic studies with 10 participants and more. Please clarify it.

7. If possible, it is better to use meta-analysis to quantify the reported associations in the results.

8. The fourth research question looks less attractive to the readers and less valuable to fall injury prevention.

9. It is necessary to mention the relationship between stair and bathroom falls and the built  environment. 

Reviewer 2 Report

This is an excellent article which has been written in a clearly and concisely. Indeed it is the first article that I have reviewed for which I have very few comments.

I have 2 comments which are as follows:

1) P2 L72 - the scope of the literature review is limited in terms of the databases searched. Obviously this cannot be changed at this point, but I would have thought that there would have been epidemiological and ergonomic literature on databases that would not be identified on Medline - perhaps this is something that could be mentioned in the limitations?

2) P17 L321 - you mention that approximately 1/3 of falls are ascribed to environmental causes. That figure is more like between 30-50% in community dwelling older people (references below). Brace 2003 quotes a figure of 44%. Perhaps just check this figure out in a couple of other publications.

Tinetti et al., 1988, Nevitt et al., 1989, Josephson et al., 1991, Nevitt et al., 1991, Weinberg and Strain, 1995, Northridge et al., 1995, Nyberg et al., 1996, CRD, 1996, Rubenstein, 1999, Hill et al., 2000, Brace et al., 2003